# Resonant out-of-phase fluorescence microscopy and remote imaging overcome spectral limitations

Jérôme Quérard[1,2], Ruikang Zhang[1,2], Zsolt Kelemen[3], Marie-Aude Plamont[1,2], Xiaojiang Xie[1,2,4], Raja Chouket[1,2], Insa Roemgens[1,2], Yulia Korepina[1,2], Samantha Albright[1,2], Eliane Ipendey[1,2], Michel Volovitch[5,6], Hanna L. Sladitschek[7], Pierre Neveu[7], Lionel Gissot[3], Arnaud Gautier [1,2], Jean-Denis Faure[3], Vincent Croquette[8], Thomas Le Saux[1,2] & Ludovic Jullien[1,2]

We present speed out-of-phase imaging after optical modulation (OPIOM), which exploits reversible photoswitchable fluorophores as fluorescent labels and combines optimized periodic illumination with phase-sensitive detection to specifically retrieve the label signal. Speed OPIOM can extract the fluorescence emission from a targeted label in the presence of spectrally interfering fluorophores and autofluorescence. Up to four fluorescent proteins exhibiting a similar green fluorescence have been distinguished in cells either sequentially or in parallel. Speed OPIOM is compatible with imaging biological processes in real time in live cells. Finally speed OPIOM is not limited to microscopy but is relevant for remote imaging as well, in particular, under ambient light. Thus, speed OPIOM has proved to enable fast and quantitative live microscopic and remote-multiplexed fluorescence imaging of biological samples while filtering out noise, interfering fluorophores, as well as ambient light.

[1] PASTEUR, Département de Chimie, École Normale Supérieure, UPMC Univ Paris 06, CNRS, PSL Research University, 75005 Paris, France. [2] Sorbonne Universités, UPMC Univ Paris 06, École Normale Supérieure, CNRS, PASTEUR, 75005 Paris, France. [3] Institut Jean-Pierre Bourgin, INRA, AgroParisTech, CNRS, Saclay Plant Sciences (SPS), Universite Paris-Saclay, 78000 Versailles, France. [4] Department of Chemistry, Southern University of Science and Technology, Shenzhen 518055, China. [5] Centre for Interdisciplinary Research in Biology (CIRB) CNRS, INSERM, Labex MemoLife, PSL Research University, Collège de France, 75005 Paris, France. [6] École Normale Supérieure, Institute of Biology at the Ecole Normale Supérieure (IBENS), CNRS, INSERM, PSL Research University, 75005 Paris, France. [7] Cell Biology and Biophysics Unit, European Molecular Biology Laboratory, 69117 Heidelberg, Germany. [8] Ecole Normale Supérieure, Département de Physique et Département de Biologie, Laboratoire de Physique Statistique, CNRS, ENS, 75005 Paris, France. Correspondence and requests for materials should be addressed to T.S. (email: Thomas.Lesaux@ens.fr) or to L.J. (email: Ludovic.Jullien@ens.fr)

In fluorescence microscopy and remote imaging, the discrimination of a fluorophore usually results from optimizing its brightness and its spectral properties[1]. Despite the widespread use of fluorescence for labeling biological samples, this approach still suffers from limitations. First, extraction of a fluorescent signal is challenging in light-scattering and autofluorescent samples. Second, spectral deconvolution of overlapping absorption and emission bands can only discriminate a few labels, falling short from the several tens needed for advanced bioimaging[2, 3].

Following our previous works exploiting kinetic parameters for selective discrimination[4–8], we have introduced Out-of-Phase Imaging after Optical Modulation (OPIOM), an imaging technique enabling selective imaging of photoactive fluorescent labels such as reversibly photoswitchable fluorescent proteins (RSFPs)[9] using their photoswitching kinetics as discriminating parameters[10, 11]. Other imaging protocols have exploited RSFP photoswitching kinetics for selective fluorescence imaging. In optical lock-in detection (OLID), the observable is the correlation coefficient between the total fluorescence emission and a reference signal from the targeted photoswitchable fluorophore over several cycles of dual-wavelength-driven photoswitching[12, 13]. In synchronously amplified fluorescence image recovery (SAFIRe), the amplitude of the fluorescence modulation generated by modulating a secondary light source depopulating dark states of the fluorophore is used to build the image[14]. As SAFIRe, OPIOM relies on a periodically modulated illumination but it exploits phase-sensitive detection and easily predictable resonance conditions involving the illumination control parameters and the RSFP photoswitching dynamics to selectively retrieve the contribution of a fluorescent probe of interest from the amplitude of the overall out-of-phase fluorescence response. The requirements for driving RSFP photoswitching at a slow rate and non-optimized extent have limited however the use of OPIOM for live fluorescence imaging[10, 11].

Here we introduce a powerful protocol, speed OPIOM, for multiplexed imaging of fluorescent labels under adverse optical conditions. Like OPIOM, speed OPIOM combines periodic illumination matched with RSFP photoswitching kinetics and phase-sensitive detection. However, speed OPIOM differs from OPIOM by a much-optimized periodic illumination modality relying on an extensive theoretical background. Speed OPIOM correspondingly enables the fast and remote imaging of an RSFP expressed in a strongly autofluorescent biological sample under ambient light and the fast, quantitative and simultaneous microscopy imaging of several spectrally overlapping RSFPs in both fixed and live cells.

## Results

**Speed OPIOM principle.** To drive RSFP photoswitching at optimal rate and extent, speed OPIOM does not use one modulated light source like the original OPIOM,[10] but two modulated light sources synchronized in antiphase at two wavelengths driving RSFP photoswitching between its bright and dark states (angular frequency of modulation $\omega = 2\pi/T$, where $T$ is the period of light modulation; mean intensities $\langle I_1 \rangle$ and $\langle I_2 \rangle$) (Fig. 1a). Tens to hundreds images are acquired for at least two periods of light excitation. The speed OPIOM algebraic signal $S$ is the out-of-phase component of the modulated fluorescence signal, which is directly retrieved without any further processing after lock-in detection with efficient noise rejection by Fourier transform of the acquired images (Fig. 1b). $S$ exhibits a sharp resonant maximum when $\langle I_1 \rangle$ and $\langle I_2 \rangle$ are tuned to maximize sensitivity of the average concentrations of the exchanging RSFP states to changes of light intensities, and $\omega$ is matched with the inverse of the RSFP photoswitching relaxation time (Supplementary Note 1). Interestingly, this maximum is twice as high as in OPIOM[10, 11]. For a given RSFP, $S$ is maximal only when the ratios $\langle I_2 \rangle/\langle I_1 \rangle$ and $\omega/\langle I_1 \rangle$ take specific values that depend only on the photochemical and kinetic properties of the RSFP, and is negligible otherwise. Each RSFP has singular photochemical and kinetic properties, and thus its own resonant conditions. Hence, tuning $\langle I_2 \rangle/\langle I_1 \rangle$ and $\omega/\langle I_1 \rangle$ to the resonant values of a specific RSFP allows to selectively image this RSFP (Fig. 1b), filtering out the contribution of non-resonant fluorophores as well as ambient light. In addition, in contrast to OPIOM, which has been limited by the thermally driven back conversion of the photo switched RSFP[10],

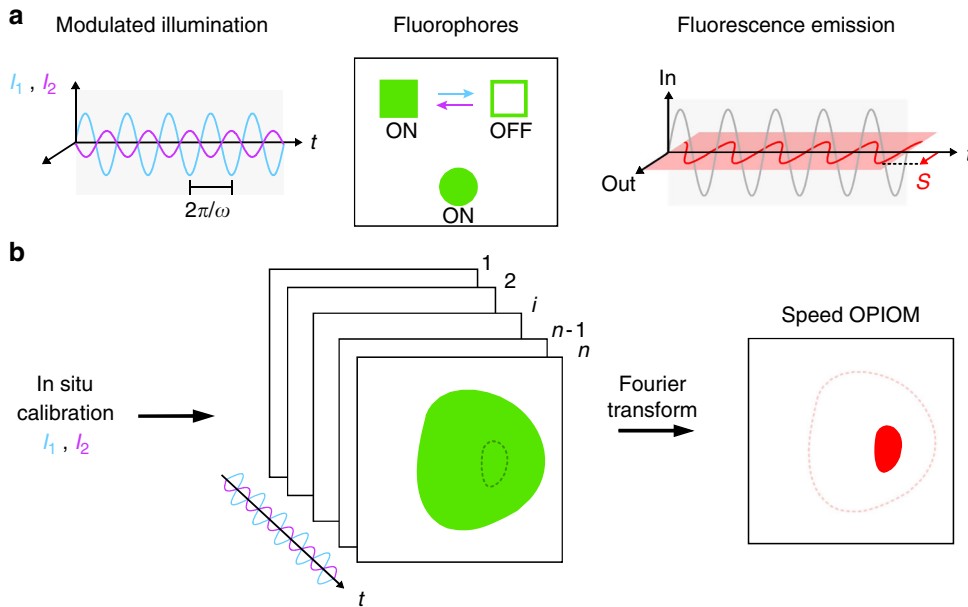

**Fig. 1** Principle of speed OPIOM. **a** A sinusoidally modulated antiphase-synchronized dual illumination generates the quadrature-delayed component $S$ (in *red*) of the fluorescence emission used for selective speed OPIOM imaging of RSFPs. **b** After in situ calibration, the fluorescence images are recorded under modulated illumination and processed to yield $S$ after Fourier transform

imaging speed can be simply increased by increasing the excitation light intensities, provided that RSFP photoconversion still obeys a two-state exchange.

**Acquisition of the speed OPIOM discrimination map**. To test the discriminative power of speed OPIOM, we used eight RFSPs: Dronpa[15], Dronpa-2[16], Dronpa-3[17], RSFastLime[16, 18], rsEFGP[19], rsEFGP2[20], Padron[18], and Kohinoor[21], which exhibit maximal emission in a narrow (505 nm; 522 nm) wavelength range[15–21] precluding spectral discrimination (Supplementary Fig. 1). Dronpa, Dronpa-2, Dronpa-3, RSFastLime, rsEFGP, and rsEFGP2 are switched to a dark or a bright state by blue and violet light, respectively (negative photochromism), whereas Padron and Kohinoor exhibit the opposite behavior (positive photochromism). We investigated the photoswitching behavior of these RSFPs and established that it obeys a two-state kinetic model at time scales longer than a few milliseconds, yielding robust photochemical and kinetic parameters with respect to environmental changes (Supplementary Note 2). Interestingly, resonant illumination conditions matching a specific RSFP could be easily determined directly on the imaging setup using a simple series of light jump experiments (Supplementary Note 3) and either purified RSFPs or fixed cells among others (Supplementary Note 2). We then computed the speed OPIOM response with varying illumination parameters for the eight assessed green RSFPs and determined that each RFSP had a distinct response maximum—analogous to an individual detection channel—in the parameter space of illumination intensities and frequencies (Fig. 2).

**Speed OPIOM is a quantitative imaging protocol**. To evidence that speed OPIOM provides quantitative information, we used a microfluidic device containing six chambers filled with Dronpa-2 solutions at five different concentrations and with the spectrally similar non-photoactive fluorescent protein EGFP. A home-built epifluorescence microscope equipped with two light emitting diodes (LEDs; Supplementary Fig. 2) driven for intensity, angular frequency of modulation, and phase was used to image the microfluidic device by tuning illumination to the Dronpa-2 resonance conditions. We recorded the processed speed OPIOM

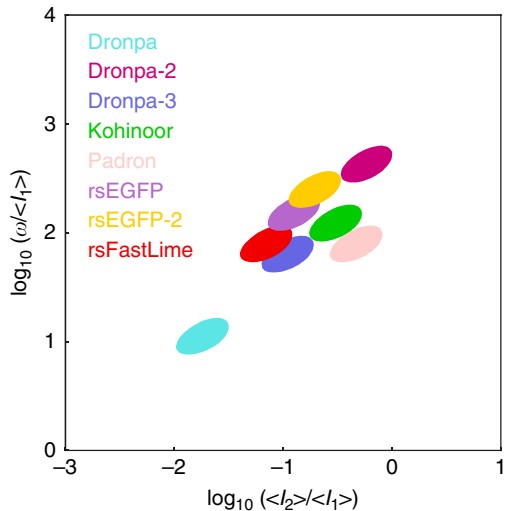

**Fig. 2** Map of RSFP discrimination with speed OPIOM. The illumination parameters $\langle I_2\rangle/\langle I_1\rangle$ and $\omega/\langle I_1\rangle$ (here in rad Ein$^{-1}$ m$^2$) can be tuned to single out the observable $S$ from multiple RSFP targets (See Supplementary Note 2 for the RSFP photochemical and kinetic parameters). The contours correspond to >95% of the maximal $S$ value with $\langle I_1\rangle = 1$ Ein s$^{-1}$ m$^{-2}$

image together with the pre-OPIOM image (built from the average fluorescence intensity) (Supplementary Fig. 3). Dronpa-2 fluorescence emission was visible in both pre-OPIOM and speed OPIOM images. In contrast, as expected from the in-phase modulation of its fluorescence emission only, no EGFP signal could be detected on the speed OPIOM image, which evidenced the anticipated selective imaging of Dronpa-2 with speed OPIOM: for EGFP and Dronpa-2 at 3 and 6 µM concentrations, respectively, pre-OPIOM and speed OPIOM images with 1:0.8 and 1:7600 intensity ratios were obtained. Furthermore, the five chambers containing Dronpa-2 exhibited relative intensities directly reflecting their concentration, which demonstrated that the speed OPIOM signal is proportional to the probe concentration as theoretically predicted.

**Speed OPIOM eliminates spectral interferences**. We then showed that speed OPIOM could selectively retrieve a targeted RSFP signal in biological samples in the presence of interfering non-photoactive fluorophores and autofluorescence (Fig. 3). In microscopy of mammalian cells expressing both Dronpa-2 and EGFP (Fig. 3a), these two proteins could not be distinguished in the pre-OPIOM image (Fig. 3b), whereas only Dronpa-2 contributed to the speed OPIOM channel (Fig. 3c and Supplementary Fig. 4). Hence, speed OPIOM efficiently suppressed the signal from spectrally interfering non-photoactive fluorophores.

**Speed OPIOM for fluorescence remote imaging under ambient light**. We then showed that speed OPIOM is not limited to microscopy but is relevant for remote imaging as well, in particular, under ambient light (Fig. 3d). Hence, we implemented speed OPIOM on a home-built fluorescence remote imaging setup (Supplementary Fig. 5) and successfully discriminated an RSFP in the highly demanding situation of remote imaging of a labeled autofluorescent plant sample under ambient light: whereas Dronpa-2-expressing *Camelina* seedlings were undistinguishable from wild-type seedlings in the pre-OPIOM image (Fig. 3e), Dronpa-2 was unambiguously revealed in the speed OPIOM channel (Fig. 3f). Thus, speed OPIOM is appropriate for selective RSFP imaging against an autofluorescent background even in ambient light.

**Speed OPIOM discriminates distinct spectrally similar RSFPs**. We first discriminated RSFPs with close resonance conditions but opposite photoswitching behaviors using the sign of their speed OPIOM signal (Fig. 4a): RSFPs with negative photochromism like Dronpa-3 contribute positively to the signal, whereas RSFPs with positive photochromism like Kohinoor contribute negatively. Thus, unlike pre-OPIOM (Fig. 4b), speed OPIOM easily distinguished Dronpa-3 and Kohinoor (Fig. 4c, d and Supplementary Fig. 6).

RFSPs possessing distinct values of the resonant illumination parameters can be imaged sequentially by adjusting the illumination conditions to the resonant values of each RSFP (Fig. 4e). Accordingly, in contrast to pre-OPIOM (Fig. 4f), the spectrally undistinguishable Dronpa and Dronpa-2 could be imaged in a microfluidic device and in mammalian cells with high-contrast enhancement in their respective speed OPIOM channels (Fig. 4g–h; Supplementary Figs. 7 and 8). Hence, in the microdevice, the pre-OPIOM and speed OPIOM images of Dronpa and Dronpa-2 at concentrations of 1 and 18 µM yielded 1:1.2 and 1:27 intensity ratios with Dronpa-2 as the target (respectively, 1:2 and 1:8 with Dronpa as the target).

In speed OPIOM, multiplexed RSFP imaging is not limited to sequential modifications of the $(\langle I_1\rangle, \langle I_2\rangle, \omega)$ set but can be parallelized (and correspondingly further accelerated) for RSFPs

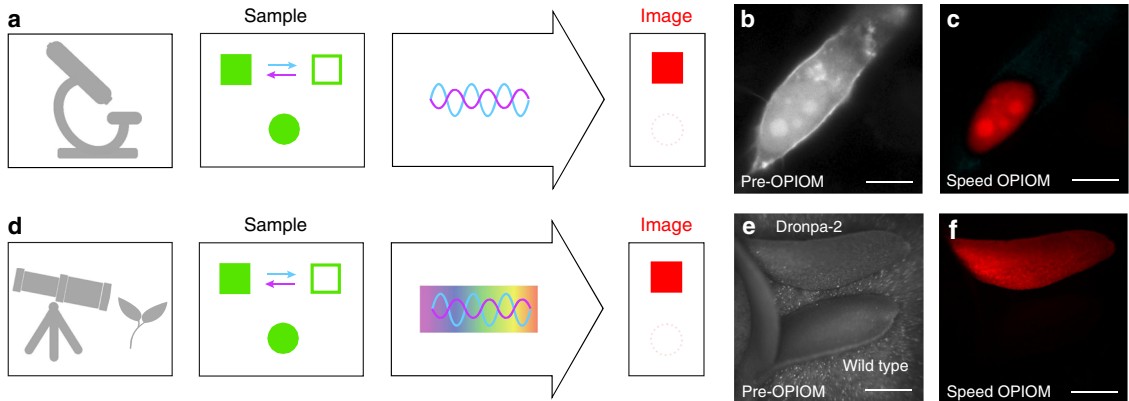

**Fig. 3** Speed OPIOM selectively retrieves an RSFP signal in the presence of spectral interferences. In contrast to pre-OPIOM (**b**, **e**), speed OPIOM microscopy (**a**) and remote (**d**) imaging selectively unveils a RSFP target (*square* in **a**, **d**, Dronpa-2 in **c**–**f**) in cells (**a**) and *Camelina* seedlings (**d**) even in the presence of a spectrally interfering fluorophore (*disk* in **a**, **d**, EGFP in **b**, **c**), autofluorescence or ambient light (**d**) and (**e**, **f**). Systems: Fixed HeLa cells expressing H2B-Dronpa-2 (at the nucleus) and Lyn11-EGFP (at the cell membrane). **b**, **c** Camelina, ubiquitously expressing Dronpa-2 or wild type (respectively *top* and *down* in **e**, **f**). The images were recorded at 37 °C (**b**, **c**) and 20 °C (**e**, **f**). *Scale bars* (μm): 10 (**b**, **c**), 875 (**e**, **f**). See Supplementary Tables 1 and 3 for the acquisition conditions

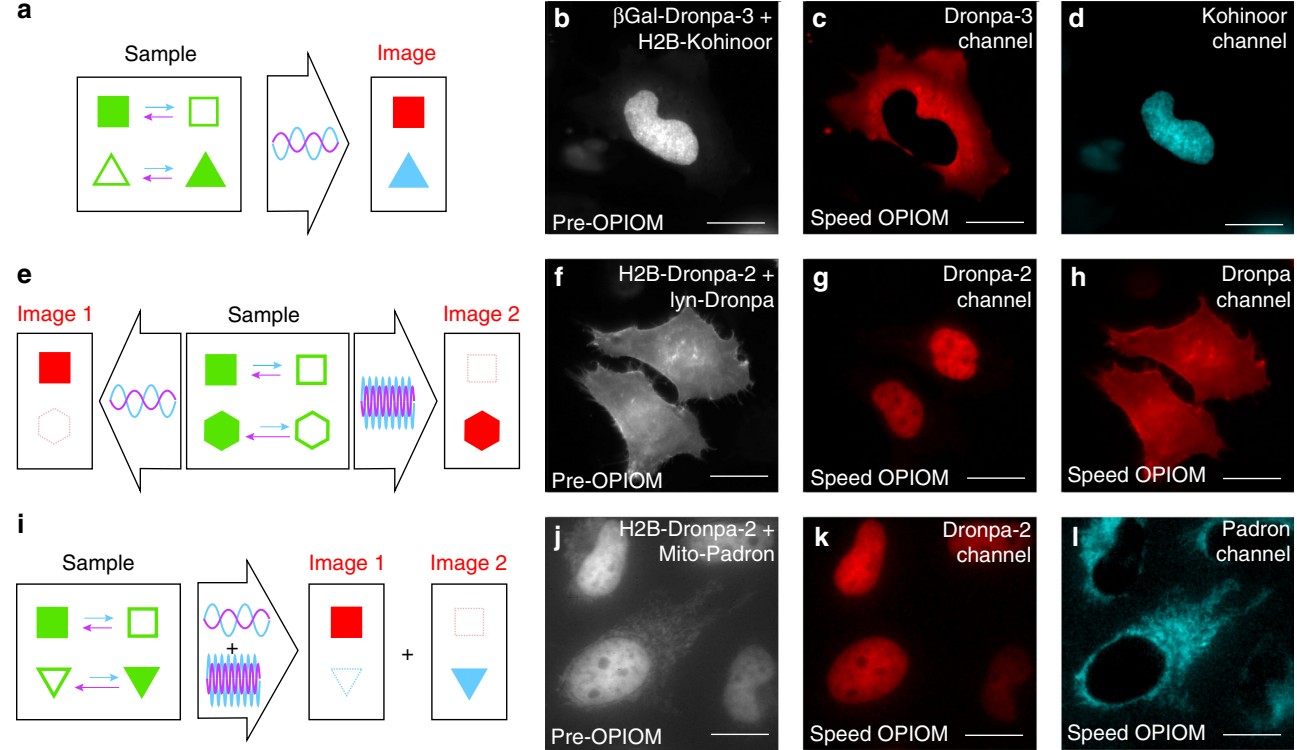

**Fig. 4** Speed OPIOM selectively discriminates RSFP signals in the presence of interfering RSFPs. Tuning the illumination parameters enables the selective imaging of RSFPs exhibiting similar resonance conditions but opposite photochromism (**a**) or distinct resonance conditions using light modulation, either sequentially at one frequency (**e**) or simultaneously at multiple frequencies (**i**). Pre-OPIOM and speed OPIOM images tuned to the resonance of Dronpa-3 and Kohinoor (**b**–**d**), Dronpa-2 (**f**, **g**) and Dronpa (**h**), or tuned to optimize orthogonality between Dronpa-2 and Padron (**j**–**l**). Systems: Fixed HeLa cells expressing βGal-Dronpa-3 and H2B-Kohinoor (**b**–**d**), H2B-Dronpa-2 and Lyn11-Dronpa (**f**–**h**), or H2B-Dronpa-2 and Mito-Padron (**j**–**l**). Localizations: βGal (cytoplasm), H2B (nucleus), Lyn11 (cell membrane). The images were recorded at 37 °C. *Scale bars*: 20 μm. See Supplementary Tables 1 and 2 for the acquisition conditions

sharing identical resonance for $<I_2>/<I_1>$ by modulating illumination at multiple resonant modulation frequencies (Fig. 4i and Supplementary Note 1). Hence, Dronpa-2 and Padron (Fig. 2) could be imaged simultaneously in a single speed OPIOM acquisition superposing two periodic illumination regimes tuned to the resonant frequency of each RFSP (Fig. 4j–l and Supplementary Fig. 9).

**Speed OPIOM for real time fluorescence imaging.** In speed OPIOM, the frequency of image acquisition can be increased by increasing $<I_1>$ and $<I_2>$ at constant value of the $<I_2>/<I_1>$ ratio. Hence, speed OPIOM gives access to imaging biological processes in live cells. To illustrate this opportunity, we monitored the nuclear translocation of ERK2 fused to Kohinoor upon epidermal growth factor (EGF) stimulation in cells with an

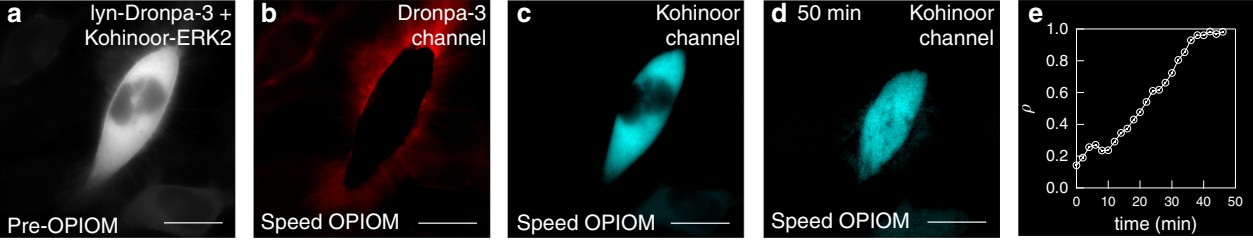

**Fig. 5** Speed OPIOM can track quantitatively fast biological processes. Live HeLa cells expressing MEK1, Kohinoor-ERK2 (initially located in the cytoplasm) and Lyn11-Dronpa-3 before (**a–c**) and after (**d**) addition of epidermal growth factor (*EGF*). The figure in **e** displays the temporal evolution of the absolute value of the ratio $\rho$ of the speed OPIOM Kohinoor signals in the nucleus and in the cytoplasm. Pre-OPIOM and speed OPIOM images tuned to the resonance of Dronpa-3 and Kohinoor (**a–e**). Localizations: Lyn11 (cell membrane). The images were recorded at 37 °C. *Scale bars*: 20 µm. See Supplementary Table 1 for the acquisition conditions

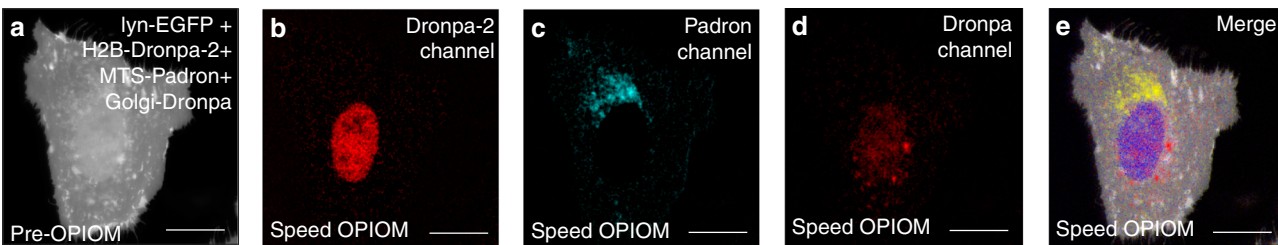

**Fig. 6** Speed OPIOM can independently image four spectrally similar fluorescent proteins without high-contrast enhancement. Pre-OPIOM (**a**) and speed OPIOM images tuned to selectively image Dronpa-2 (**b**), Padron (**c**), and Dronpa (**d**). In **e**, overlay between the pre-OPIOM image from the Dronpa-2 acquisition and speed OPIOM images collected in (**b–d**). Systems: Fixed U2OS cells expressing H2B-Dronpa-2, Mito-Padron, Dronpa-GTS, and Lyn11-EGFP. Localizations: H2B (nucleus), Lyn11 (cell membrane), Mito (mitochondria), GTS (Golgi). The images were recorded at 37 °C. *Scale bars*: 20 µm. See Supplementary Table 1 for the acquisition conditions

interfering Dronpa-3 membrane label (Fig. 5a–d). Speed OPIOM allowed following the rapid Kohinoor-ERK2 nuclear transloca-tion (Fig. 5e), demonstrating that speed OPIOM can be used to track quantitatively fast biological processes.

**Speed OPIOM for multiplexed fluorescence imaging**. Finally we demonstrated that speed OPIOM could overcome spectral lim-itations for multiplexed imaging. Hence, Dronpa, Dronpa-2, and Padron could be first imaged in a microdevice with high-contrast enhancement using three acquisitions involving different ($<I_1>$, $<I_2>$, $\omega$) sets (Supplementary Note 5). They were subsequently imaged in cells (Fig. 6a–d and Supplementary Fig. 10) to build a merged image unambiguously displaying four spectrally similar fluorescent proteins—EGFP, Dronpa, Dronpa-2, and Padron (Fig. 6e).

## Discussion
In this paper, we used RSFPs to show that speed OPIOM is a powerful approach for multiplexed fluorescence imaging against a background of spectral interferences. Speed OPIOM efficiently filters out autofluorescence, scattered and ambient light, opening up great prospects for remote fluorescence imaging within highly autofluorescent biological samples under broad daylight. Speed OPIOM has enabled live imaging of biological processes at the second time scale but is compatible with a speed of image acquisition of up to 50 Hz by adjusting the illumination light intensities. Eventually, as speed OPIOM only requires the mod-ulation of illumination light and phase-sensitive detection of the fluorescence, such features could be easily implemented on existing fluorescence imaging setups in addition to optical filters for spectral discrimination.

Speed OPIOM distinguishes spectrally similar RSFPs (but other photoswitchable fluorophores[22–25] would be relevant as

well) by leveraging their photochemical and kinetic character-istics. Thanks to its phase-sensitive detection scheme, speed OPIOM is both reference-free and band-pass-selective thereby overcoming some limitations encountered with SAFIRe and OLID (Supplementary Note 5). Despite intrinsic limitations associated to noise (Supplementary Note 6) and precision on phase retrieval (currently $3 \times 10^{-3}$ rad), speed OPIOM imaging can typically enhance contrast of photoswitchable fluorophores against non-photoactive spectrally interfering fluorophores or ambient light by a $10^2$–$10^3$ factor. Eventually, speed OPIOM application will be facilitated by the ever-expanded number of RSFPs developed for super-resolution microscopy; by requiring one order of magnitude departure of the photochemical proper-ties for efficient discrimination, we estimate that speed OPIOM should soon give access to multiplexed imaging of five to ten distinct RSFPs. Hence, speed OPIOM should considerably expand the scope of fluorescence microscopy and remote imaging at video rate.

## Methods
**Reversibly photoswitchable fluorescent proteins**. Most of the RSFPs used in this study take part to the Dronpa[15] family. Dronpa-2[16] and rsFastLime[16] contain only one mutation M159T and V157G respectively whereas Dronpa-3[17] has two mutations: V157I/M159A. Padron[18] carries eight mutations: T59M, V60A, N94I, P141L, G155S, V157G, M159Y, and F190S. Kohinoor[21] is a Padron variant con-taining seven mutations (according to the Padron sequence): N102I, L141P, F173S, S190D, D192V, K202R, and E218G. rsEGFP1 (EGFP(T65A, A206K)[19] and rsEGFP2 (EGFP(T65A, Q69L, V163S, A206K)[20] have been obtained from EGFP and contained two and four mutations, respectively.

**Plasmids**. The plasmids for bacterial expression of Dronpa-2[10], Dronpa-3[10], Dronpa[26], Kohinoor (pRSET-Kohinoor)[21], rsEGFP1 (pBAD-rsEGFP)[19], and rsEGFP2 (pBAD-rsEGFP2)[20] carrying an N-terminal hexahistidine tag were pre-viously described. The plasmid for bacterial expression of rsFastLime was obtained by introducing the V157G mutation in that of Dronpa, using the QuickChange Site-Directed Mutagenesis Kit (Stratagene).

The plasmid pcDNA3-Kohinoor-H2B for mammalian expression of H2B-Kohinoor was previously described by Nagai and coworkers[21]. The plasmid for mammalian expression of Lyn11-EGFP (with a N-terminal MGCIKSKGKDSAGGGS sequence for membrane targeting) was previously described[10]. The plasmid pJWCAG90 for mammalian expression of MEK1 was previously described by Gautier et al[27]. The plasmid for mammalian expression of Dronpa and Dronpa-3 with a N-terminal MGCIKSKGKDSAGGGS sequence (Lyn11) for membrane targeting (Lyn11-Dronpa and Lyn11-Dronpa-3) were obtained by insertion of the genes encoding Lyn11-Dronpa and Lyn11-Dronpa-3 in the pIRES vector (Invitrogen) using *Bgl*II and *Not*I restriction sites. The plasmid for mammalian expression of Dronpa-2 fused at the C terminal of the histone H2B (H2B-Dronpa-2) was obtained by insertion of the genes encoding H2B-Dronpa-2 in the pIRES vector (Invitrogen) using *Bgl*II and *Not*I restriction sites. The plasmid for mammalian expression of Dronpa-3 fused to the C terminal of *β*-galactosidase (*β*Gal-Dronpa-3) was obtained by inserting the genes encoding *β*Gal-Dronpa-3 in the pIRES vector (Invitrogen) using *Bgl*II and *Not*I restriction sites. The plasmid for mammalian expression of Kohinoor fused to the N terminal of ERK2 (Kohinoor-ERK2) was obtained by inserting the genes encoding Kohinoor-ERK2 in the pEGFP-C vector (Clontech) using *Nhe*I and *Kpn*I restriction sites. The plasmids for the mammalian expression of EGFP fused to the membrane anchoring (lyn) peptide from the mouse tyrosine-protein kinase Lyn, Dronpa fused to a 25 aa Golgi-targeting signal (GTS) from human Golgi phosphoprotein 2 (GOLPH2), Padron fused to the N-terminal mitochondrial targeting signal (MTS) of either COX4 or COX8 and Dronpa-2 fused to human H2B for chromosomal location were assembled using MXS chaining[28]. We sequentially inserted a CMV promoter (or sCMV for Padron), the lyn (MGCIKSKRKDVEN), MTS (MLSLRQSIRFFKPATRTLCSSRYLL or MSVLTPLLLRGLTGSARRLPVPRAKIHSL), GTS (MKSPPLVLAALVACIIVLGFNYWIA) peptide tags or the H2B coding sequence to plasmids containing the coding sequences of EGFP, Dronpa, Padron, or Dronpa-2, respectively, coupled to the bovine growth hormone (BGHpA, alternatively SV40LpA for Padron) polyadenylation signal. The expression blocks CMV::GTS-Dronpa-BGHpA and CMV::Lyn-eGFP-BGHpA were further combined to give a bicistronic plasmid. CMV::MTS-Padron-BGHpA and CMV::H2B-Dronpa-2-BGHpA blocks were similarly combined.

The Camelina expression vector expressing p35S::Dronpa2 construct was achieved as follow. The gateway-cassette from pUBN-GFP-Dest[29] amplified by PCR using primers 5′-tgacgcgtaaggggatctagaatcacaagtttgtacaaaaagctg-3′ and 5′-ttaatcacactcaccatctcgaggatcaccactttgtacaagaaag-3′ and Dronpa-2 coding sequence amplified from Dronpa-2pDONR207 with primers 5′-atggtgagtgtgatta aaccagaca-3′ and 5′-ggctgcggccgcctcgactacttggcctgcctcggcagctcag-3′ were cloned in one step in *Bam*HI *Xho*I-digested pBinGlyRed vector[30], creating pD2. The doubled CaMV 35 S promoter was amplified from pMDC83[31] with primers 5′-ggggacaag tttgtacaaaaaagcaggcttccagtgccaagcttggcgtgcct-3′ and 5′-ggggaccactttgtacaagaaagct gggtcgtcgaggtcctctccaaatgaaatg-3′, recombined into pDONR207 (Invitrogen) and eventually recombined into pD2.

## Protein production and purification.

The plasmids expressing Dronpa, Dronpa-2, Dronpa-3, rsFastLime, Padron, and Kohinoor carrying an N-terminal hex-ahistidine tag were transformed in *E. coli* DH10B strain. Cells were grown in Terrific Broth (TB). Expression was induced by addition of isopropyl *β*-D-1-thio-galactopyranoside (IPTG) to a final concentration of 1 mM at $OD_{600}$ 0.6. Cells were harvested after 16 h of expression and lysed by sonication in Lysis buffer (30/40 mM imidazole, 50 mM TRIS/HCl at pH 7.5, 400 µM 4-(2-Aminoethyl) benzenesulfonyl fluoride hydrochloride, 5 mg ml$^{-1}$ DNAse, 5 mM $MgCl_2$ and 1 mM dithiothreitol). Insoluble material was removed by centrifugation and the soluble protein extract was batch adsorbed onto Ni-NTA agarose resin (Thermofisher). The protein loaded Ni-NTA column was washed with 20 column volumes of 50 mM TRIS/HCl pH 7.5, 20 mM imidazole, 150 mM NaCl. Bound protein was eluted in 50 mM TRIS/HCl pH 7.5, 500 mM imidazole, 150 mM NaCl. Protein fractions were dialyzed on cassette Slide-A-Lyzer Dialysis Cassettes (Thermofisher) against 50 mM TRIS/$H_2SO_4$ pH 8.

The plasmids pBAD-rsEGFP and pBAD-rsEGFP2 were transformed in E. coli BL21 cells (Merck Millipore). Cells were grown in Lysogeny Broth (LB) supplemented with 0.1% glucose w/v. Expression was induced by addition of arabinose to 0.2% w/v at OD600 0.4. Cells were collected after 4 h of expression and lysed by sonication in 50 mM PBS pH 7.4, 150 mM NaCl and a cocktail of protease inhibitors (Sigma Aldrich). Insoluble material was removed by centrifugation and the soluble protein extract was batch adsorbed onto Ni-NTA agarose resin (Thermofisher). The protein loaded Ni-NTA column was washed with 20 column volumes of 50 mM PBS pH 7.4, 150 mM NaCl, 20 mM imidazole. Bound protein was eluted in 50 mM PBS pH 7.4, 150 mM NaCl, 0.5 M imidazole. Protein fractions were dialyzed on cassette Slide-A-Lyzer Dialysis Cassettes (Thermofisher) against 50 mM PBS pH 7.4, 150 mM NaCl.

## Mammalian cell culture and transfection.

HeLa and U2OS cells were grown at 37°C in 5% $CO_2$ atmosphere in DMEM with GlutaMAX-1 complemented with 10% fetal bovine serum (FBS) and 1% penicillin / streptomycin. Cells were transiently transfected with Genejuice (Merck) according to the manufacturer's protocol. Cells were washed with Dulbecco's phosphate-buffered saline (DPBS) and fixed with 2% paraformaldehyde (PFA) solution.

## Camelina sativa transformation and growth.

*Camelina sativa* (cv Celine) was transformed following an improved method of the traditional *Arabidopsis* floral-dip method and transgenic were selected as described previously[32]. Seeds were sown on water-soaked paper and grow for 7 days in a growth chamber under cycles of 16 h light at 22°C/8 h dark at 16°C.

## Reagents and solutions.

All solutions were made up using purified water (Direct-Q 5 apparatus; Millipore, Billerica, MA). The Britton-Robinson buffer was prepared from acetic acid: 4 mM; phosphoric acid: 4 mM; AMPSO: 4 mM, NaCl: 150 mM. All experiments conducted to determine the photophysical and photo-chemical properties of the purified proteins were performed in Britton-Robinson buffer (pH 7.5) at 37 °C, using protein concentrations of 10 µM, unless stated otherwise.

## pH measurements.

By assimilating activity and concentration, the proton con-centration was directly measured after calibration of the pH meter (Standard pH meter PHM210, Radiometer Analytical equipped with a Radiometer Analytical PHC3359-8 combination pH electrode (Hach, Loveland, CO).

## Spectroscopic instruments.

Absorption spectra of proteins were recorded at 37 °C on a Cary 300 UV/Vis spectrophotometer (Agilent Technologies, Santa Clara, CA), equipped with a Peltier 1 × 1 thermostated cell holder (Agilent Technologies). The samples were placed in 55 µl quartz cuvettes (0.3 cm × 0.3 cm light path; Hellma Optics, Jena, Germany). Fluorescence measurements used for kinetic analysis were acquired on a LPS 220 spectrofluorometer (PTI, Monmouth Junction, NJ), equipped with a TLC50 cuvette holder (Quantum Northwest, Liberty Lake, WA) thermoregulated at 37 °C. Light intensities were controlled by varying the current on two LED light sources. The first one (LXZ1-PB01 from Philips Lumileds, San Jose, CA) was filtered at 480 ± 20 nm (HQ 480-40 from Chroma Technology Corp, Rockingham, VT) whereas the second one (LHUV-0405, Philips Lumileds) was filtered at 405 ± 20 nm (F405-40; Semrock, Rodchester, NY). The LEDs were supplied by a DC4100 LED driver (Thorlabs, Newton, NJ). The two light sources were collimated with ACL2520U condenser lenses (Thorlabs) and beams were next combined thanks to a dichroic filter (T425LPXR, Chroma Technology Corp). Photon fluxes were measured with a Nova II powermeter (Laser Measurement Instruments).

## Microfluidic devices.

Each microdevice was composed of a circular glass coverslip (0.17 mm thick, 40 mm diameter; Menzel-Glaser, Braunschweig, Germany) and a PDMS stamp (RTV615; General Electrics, Fairfield, CT) including either four 400 µm × 400 µm × 20 µm square chambers separated by 150 µm × 20 µm walls or six 250 µm × 125 µm × 20 µm chambers separated by 100 µm × 20 µm walls. Each chamber was connected to a sample reservoir punched in the PDMS stamp through a 40 µm × 20 µm channel. Before assembly, the coverslip and the PDMS stamp were rinsed with ethanol and dried under a nitrogen flow. The bottom glass surface of the microdevice was placed on a 0.4 mm thick copper disk in which a 8 mm hole had been opened for further observation with the objective. To fill the micro-chambers with appropriate solutions, the air dissolved in the PDMS was pumped for 3 min at 50 mbar at room temperature and sample solutions were added to each reservoir, which resulted in the autonomous and controlled loading of the device.

## Microscopy epifluorescence setup.

We used a home-built epifluorescence microscopy setup (Supplementary Fig. 2). The samples were illuminated using a LXZ1-PB01 LED (Philips Lumileds) filtered at 480 ± 20 nm (F480-40; Semrock, Rochester, NY) and a LHUV-0405 (Philips Lumileds) LED filtered at 405 ± 20 nm (F405-40; Semrock, Rochester, NY) as light sources. Each LED was supplied by a LED driver (LEDD1B, Thorlabs, Newton, NJ) and modulated synchronously to each other by a waveform generator (33612 A, Keysight Technologies). A lens (ACL2520U; Thorlabs, Newton, NJ, $f = 20$ mm) was placed just after each diode to collimate the light sources. The two light beams were next combined thanks to a dichroic mirror (T425LPXR, Chroma, Bellows Falls, VT) and a second pair of lenses was used to focus the light at the back focal plane of the objective after being reflected by the dichroic filter (Di-FF506, Semrock, Rochester, NY). Fluorescence images at 525 ± 15 nm (F525-30; Semrock, Rochester, NY) were acquired for the microdevices with a 10 × fluar (NA 0.5, Carl Zeiss AG, Feldbach, Switzerland) objective and for cell imaging with a 60 × UPlanApo (NA 1.2, Olympus Cor-poration, Tokyo, Japan) objective. Objectives were mounted on a home-built microscope equipped with a Luca-R CCD camera (Andor Technology, Belfast, UK). The bottom surface of the imaged sample was placed on a 0.4 mm thick copper disk in which a hole of 8 mm in diameter had been opened for observation with the objective. This metal holder was itself mounted on an aluminum block thermostated at 37 ± 0.2 °C with two thermoelectric Peltier devices (CP 1.0-63-05 L-RTV; Melcor, Trenton, NJ). The stage temperature was monitored with a TCS610 thermistor (Wavelength Electronics, Bozeman, MT) and the feedback loop was driven by a MPT10000 temperature controller (Wavelength Electronics, Bozeman, MT). The average excitation intensities were calibrated in imaging conditions by measuring the relaxation times associated to the photoisomerizations of Dronpa-2 containing samples upon irradiation with one or two wavelengths

(Supplementary Note 3). Triggering of the camera acquisition was synchronized with the onset of the periodic excitation light (using the option "External start" in the Solis software, Andor Technology). A fixed phase delay $\phi_{acq}$ between the dates of camera recording and light excitation was observed accordingly (due to integration of the signal over the exposure time and the triggering of the camera). This phase delay has been previously calibrated using a microfluidic device filled with Fluorescein or using cells expressing EGFP.[10] the $I_F^{1,out}$ and $I_F^{1,in}$ first-order in- and out-of-phase components of temporal evolution of Fluorescein (or EGFP) emission were extracted (see the Matlab code given in Supplementary Note 4) and $\phi_{acq}$ was computed as $\phi_{acq} = \arctan(I_F^{1,out}/I_F^{1,in})$. This value was then used for further image analysis.

**Setup for remote fluorescence imaging**. We used a home-built remote sensing setup (Supplementary Fig. 5) for macro imaging. In addition to the two LED sources at 480 and 405 nm used in the microscopy setup, we introduced a LXZ1-PB01 LED (Philips Lumiled) filtered at $550 \pm 15$ nm (ET550/15×; Chroma, Bellows Falls, VT) to possibly excite red fluorescent emitters (for example DsRed). The three light beams were combined by three dichroic mirrors (T425LPXR, T505LPXR, 59004bs; Chroma, Bellows Falls, VT). A beam expander system consisted of one divergent lens (ACN254-040-A, $f = -40$ mm, Thorlabs, Newton, NJ) and two convergent lenses (AC508-100-A, $f = 100$ mm, Thorlabs, Newton, NJ) has been used so that the light is clearly focused at a distance of up to 130 mm away where the sample is placed. With this setup, we can detect both green (525 nm) and red (600 nm) fluorescence emissions. A specially designed objective with three elements has been correspondingly used to correct for the chromatism of these two channels. The sample image is acquired by a color CMOS camera (UI-3060CP, iDS Imaging Development Systems GmbH, Obersulm, DE). The sample is positioned on a z-axis translation stage, which allows us to find the focus. The average excitation intensities have been calibrated by measuring the relaxation time of the Dronpa-2 photoisomerization. The modulated signals are generated by the computer and applied to the LEDs through an Arduino microcontroller. The acquisition of the camera is synchronized with the modulated excitation light. A series of images of the red (DsRed) and the green (Dronpa-2) channel can be separately acquired by the camera and sent back to the computer for later processing. The I/O control system and the image processing are accomplished with a home-made software coded in C.

**Softwares**. Data treatment, image analysis, and theoretical computations were performed using Igor Pro (WaveMetrics), MATLAB (The MathWorks), and Gnuplot and Mathematica (Wolfram Research) softwares.

**Data availability**. Data available on request from the authors.

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

## Acknowledgements

We thank A. Miyawaki for providing the Dronpa-3 cDNA; S. Jakobs for providing the Dronpa-2, rsEGFP, and rsEGFP2 cDNAs; A. Espagne for providing the Dronpa, Padron, and rsFastLime cDNAs; and T. Nagai for providing the Kohinoor cDNA. We are also indebted to E. Billon-Denis (for help in preparing RSFPs), and O. Mesdjan and J. Fattaciolli (for access to clean rooms to fabricate the microdevices). This work was supported by the ANR (France BioImaging—ANR-10-INBS-04, Morphoscope2—ANR-11-EQPX-0029), the SATT Lutech (OPIOM), the Fondation de la Recherche Médicale (FRM DEI201512440), the LabEx Saclay Plant Sciences-SPS (ANR-10-LABX-0040-SPS) from the "Investments for the Future" program (ref. ANR-11-IDEX-0003-02), PSL Research University (ANR-10-IDEX-0001-02 PSL, SuperLINE project), and the Domaine d'Intérêt Majeur Analytics de la Région Ile de France (DREAM).

## Author contributions

L.J. and T.L.S. conceived the speed OPIOM imaging protocol and supervised the project. J.Q., R.Z., X.X., R.C., I.R., Y.K., S.A., V.C., T.L.S., and L.J. designed and performed theoretical calculations and/or physicochemical experiments. Z.K., M.A.P.,

E.I., M.V., H.L.S., P.N., L.G., A.G., and J.-D.F. designed and prepared the biological samples.

## Additional information

**Competing interests:** The authors declare no competing financial interests.

**Change history:** A correction to this article has been published and is linked from the HTML version of this paper.

