## [Peer Review File · Nature Communications]

Reviewers' comments:

Reviewer #1 (Remarks to the Author):

I believe the work is solid and should be considered for publication in Nature Communications providing the authors address the following points.

The strength of the study lies in the application of a modulation approach to isolate specific fluorescence signals from genetically-encoded optical switch probes (2-state fluorescent proteins), and their ability to isolate these signals against large background sources (autofluorescence or ambient light). The authors have argued that Speed OPIOM could overcome fundamental limitations of conventional illumination-detection approaches to improve the signal contrast of suitably labeled optical switch probes in SRI-microscopy and live cell imaging in natural tissue.

Abstract:

The abstract is somewhat technical, and it does not convey an intensity of progress and excitement that would pique the interest of microscopists and cell biologists. I recommend the authors work to remedy this problem.

Main text:

"Despite the widespreadmajor limitations. "
Surely fluorescence is not used to label biological samples.

The authors should ask themselves whether if they have been fair in their account of OLID and SAFIR, and the influences of these earlier modulated fluorescence microscopies on the development of their imaging technique (Speed OPIOM)?

"Unlike other imaging protocolsis reference-free.."

This statement is not strictly true. In OLID one can also isolate a modulated fluorescence signal in the sample from other un-modulated sources using an "internal" reference - this signal is inherent in the image series and in this way is no different from the title method.

I would also apply this miscategorization to SAFIR, as here one the time-dependent change of the fluorescence signal is stochastic External references were used in the earlier techniques to validate specific organic and inorganic probes as high-fidelity probes for optical switching. Once validated, one can apply an "internal" reference to isolate the modulated component of tyhe signal from the samples ie, regions of the labeled sample that show the highest modulation of optical switching. Follow on articles were developed that detailed signal processing in both techniques

Qualitatively, the data presented by the authors suggest that Speed OPIOM achieves similar contrast enhancements as OLID and SAFIR. I think the authors need to comment on this, as the only reference they have made to OLID and SAFIR appears to be a negative one.

The authors should cite studies that have used small organic probes that have been used in vitro to generate high-contrast images in solution and in living cells. The . reason for this inclusion is to highlight the well-known fact that optical switching between the 2-states of a fluorescent protein is generally slow. These switching rates are much faster for rationally designed optical switch probes, including red-shifted NitroBIPS - these probes may well enhance and extend applications of Speed OPIOM to living systems.

other points

What was the power of the light source at the sample for studies detailed in the manuscript - how do they compare to conventional imaging and OLID/SAFIR?
Can Speed OPIOM be used to isolate rare events such as low efficient FRET between appropriate donor and acceptor probes?

Reviewer #2 (Remarks to the Author):

The manuscript by Querard and coworkers describes an extension of a method previously presented by the same group (Ref. 4). The method (termed Speed OPIOM) takes advantage of the differences in the switching kinetics between different photoswitchable fluorophores to discriminate between them despite their overlapping emission spectra. In the previous work, switching was induced by a monochromatic modulated optical field. Here, a bichromatic field is introduced, enabling better control and more rapid switching between the two states. Speed OPIOM is demonstrated in various scenarios of imaging and is shown to yield excellent discrimination between up to 8 different fluorophores. Overall, the manuscript is very well written, and the presented experimental results are very impressive. In terms of quality and scope, the manuscript can certainly be accepted to Nature Communications, provided that the following (rather minor) points are addressed:

1. The difference between OPIOM and Speed OPIOM is not sufficiently well explained in the introduction. The difference between a monochromatic and a bichromatic excitation field and the assumptions (some of which are presented in the SI) should be clearly stated in the introduction.
2. The ratio Ω/I_1 is confusing. These are two quantities of different units, hence cannot be directly related to one another. In fact, they are related through the absorption cross section of the molecule. The authors should use a proper notation here.
3. The term "radial frequency" is confusing. It should be "modulation frequency".
4. The phrase "without any cross talk" is repeated numerous times in the text. This is generally an incorrect and non-quantitative statement. It would be much better if the authors discuss cross talk in terms of a reduction factor (which can be very large but is still nonzero).
5. The authors should add some discussion on crosstalk through detector saturation and through enhanced noise. In fact, all modulation based techniques are limited by these. Even a general discussion on the maximal magnitude of signal to background and on the practical number of independent channels realizable would do.
6. It would be good if the authors added to the SI a graph including all the emission spectra of the fluorescent proteins used in this work.

Answers to the reviewers' comments

Reviewer #1 (Remarks to the Author)

1. I believe the work is solid and should be considered for publication in Nature Communications providing the authors address the following points.

The strength of the study lies in the application of a modulation approach to isolate specific fluorescence signals from genetically-encoded optical switch probes (2-state fluorescent proteins), and their ability to isolate these signals against large background sources (autofluorescence or ambient light). The authors have argued that Speed OPIOM could overcome fundamental limitations of conventional illumination-detection approaches to improve the signal contrast of suitably labeled optical switch probes in SRI-microscopy and live cell imaging in natural tissue.

We are thankful to the reviewer to consider that our work could be published in Nature Communications.

2. Abstract: The abstract is somewhat technical, and it does not convey an intensity of progress and excitement that would pique the interest of microscopists and cell biologists. I recommend the authors work to remedy this problem.

We modified the abstract in order to address the reviewer's concern.

3. Main text: "Despite the widespreadmajor limitations." Surely fluorescence is not used to label biological samples.

We modified this sentence in order to address the reviewer's concern.

4. The authors should ask themselves whether if they have been fair in their account of OLID and SAFIR, and the influences of these earlier modulated fluorescence microscopies on the development of their imaging technique (Speed OPIOM)?

The introductory paragraph of our initial manuscript was concise. Hence we explained in one sentence the singularity of Speed OPIOM with respect to the original versions of OLID and SAFIR. This conciseness (which has been applied to precise the limitations of our OPIOM as well...) may explain the feeling of the reviewer. In the introduction of the revised manuscript, we explain what are OLID, SAFIR, and OPIOM and make more explicit how Speed OPIOM differs from these three protocols.

Although OLID, SAFIR, and OPIOM all rely on discrimination based on kinetics of exchanges between various photophysical states of photoswitchable fluorescent probes, our OPIOM development originates from our own previous work relying on kinetics of chemical reactions for highly selective discrimination of a targeted species in complex mixtures. This origin is now made explicit in the revised version of the Main Text: beyond our numerous unquoted

theoretical developments (which have started in 2000), we introduced four references with three of them including experimental validations.

5. *"Unlike other imaging protocolsis reference-free." This statement is not strictly true. In OLID one can also isolate a modulated fluorescence signal in the sample from other un-modulated sources using an "internal" reference - this signal is inherent in the image series and in this way is no different from the title method.*

OLID requires a reference containing the targeted photoswitchable fluorophore, which can be either endogenous, be added to the observed sample, or be acquired independently in a previous experiment. But a reference waveform is necessary in order to build the OLID image.

6. *I would also apply this miscategorization to SAFIR, as here one the time-dependent change of the fluorescence signal is stochastic.*

There was no miscategorization. In the original manuscript, we argued that in contrast to OLID and SAFIRe, OPIOM was both reference-free **and** band-pass-selective. SAFIRe is reference-free but is not band-pass-selective (whereas OLID is band-pass-selective but not reference-free) In the revised version of the manuscript, we explain in more details how OPIOM and Speed OPIOM differ from OLID and SAFIRe.

7. *External references were used in the earlier techniques to validate specific organic and inorganic probes as high-fidelity probes for optical switching. Once validated, one can apply an "internal" reference to isolate the modulated component of the signal from the samples ie, regions of the labeled sample that show the highest modulation of optical switching. Follow on articles were developed that detailed signal processing in both techniques.*

See 5. In addition, we introduced follow on articles in the revised manuscript.

8. *Qualitatively, the data presented by the authors suggest that Speed OPIOM achieves similar contrast enhancements as OLID and SAFIR. I think the authors need to comment on this, as the only reference they have made to OLID and SAFIR appears to be a negative one.*

Following the recommendation of the reviewer, we compared the results of the application of the Speed OPIOM, SAFIRe, and OLID imaging protocols under resonant modulated illumination to target Dronpa, Dronpa-2, or Padron. The results are now discussed in a supplementary note (Supplementary Note 5) of the Supporting Information. They led us to summarize the advantages (as well as the limitations) of Speed OPIOM in the discussion of the revised manuscript.

9. *The authors should cite studies that have used small organic probes that have been used in vitro to generate high-contrast images in solution and in living cells. The reason for this inclusion is to highlight the well-known fact that optical switching between the 2-states of a fluorescent protein is*

generally slow. These switching rates are much faster for rationally designed optical switch probes, including red-shifted NitroBIPS - these probes may well enhance and extend applications of Speed OPIOM to living systems.

We perfectly agree with the reviewer that small photoswitchable fluorophores could enhance and extend applications of Speed OPIOM for fluorescence imaging. This point has been made clear in the discussion of the revised manuscript.

10. What was the power of the light source at the sample for studies detailed in the manuscript - how do they compare to conventional imaging and OLID/SAFIR?

Speed OPIOM does not require fixing the power of the light sources at resonance but only the ratio of the average powers of both light sources. To increase the excitation light intensities enables us to increase the imaging speed provided that photoconversion of the photoswitchable fluorophore still obeys a two-state exchange.

The power of the light sources used in Speed OPIOM and OLID are essentially the same in wide-field microscopy; from Marriott et al, we noticed that Dronpa photoswitching at 488 nm is in the 20 s range, whereas we obtained 10 s in our Speed OPIOM experiments. In contrast, SAFIRE has used various light powers depending on the instrumental configuration. Considering the SAFIRE principle, we estimate that Speed OPIOM and SAFIRE (under optimal conditions) would use similar light powers in epifluorescence microscopy.

11. Can Speed OPIOM be used to isolate rare events such as low efficient FRET between appropriate donor and acceptor probes?

We are thankful to the suggestion of the reviewer. Indeed Speed OPIOM should be able to facilitate discrimination between FRET and direct excitation beyond any spectral consideration

[redacted]

However, so far, we did not yet use Speed OPIOM in FRET Donor-Acceptor pairs in which (at least) one member is reversibly photoswitchable.

Reviewer #2 (Remarks to the Author)

1. The manuscript by Querard and coworkers describes an extension of a method previously presented by the same group (Ref. 4). The method (termed Speed OPIOM) takes advantage of the differences in the switching kinetics between different photoswitchable fluorophores to discriminate between them despite their overlapping emission spectra. In the previous work, switching was induced by a monochromatic modulated optical field. Here, a bichromatic field is introduced, enabling better control and more rapid switching between the two states. Speed OPIOM is

demonstrated in various scenarios of imaging and is shown to yield excellent discrimination between up to 8 different fluorophores. Overall, the manuscript is very well written, and the presented experimental results are very impressive. In terms of quality and scope, the manuscript can certainly be accepted to Nature Communications, provided that the following (rather minor) points are addressed.

We are thankful to the reviewer to consider that our work could be published in Nature Communications.

2. The difference between OPIOM and Speed OPIOM is not sufficiently well explained in the introduction. The difference between a monochromatic and a bichromatic excitation field and the assumptions (some of which are presented in the SI) should be clearly stated in the introduction.

The introductory paragraph of our initial manuscript was concise. In the revised manuscript, we explain in more details how Speed OPIOM differs from OPIOM and what are the assumptions underlying Speed OPIOM use.

3. The ratio Ω/I is confusing. These are two quantities of different units, hence cannot be directly related to one another. In fact, they are related through the absorption cross section of the molecule. The authors should use a proper notation here.

We agree with the reviewer that using the ω/I_1^0 ratio as a control parameter of illumination lacks elegance. In fact the two-state dynamic model involve two adimensional control parameters: $\theta = \omega\tau_{12}^0$ (defined in Eq. (57) of the SI) and K_{12}^0 defined in Eq.(19) of the SI). However, all RSFPs give rise to identical resonance conditions and response maps in the (θ, K_{12}^0) space. In addition, at the end, one has to come back to a dimensioned space involving two control parameters in order to discriminate distinct RSFPs possessing different resonance conditions with Speed OPIOM: we adopted I_2^0/I_1^0 and ω/I_1^0 , which simply emerge from the resonance conditions given in Eqs.(100,101). Since this article is primarily intended to end users (in particular biologists), we decided to use at once the dimensioned space for our figures. In fact, we presently finalize another manuscript on the response of reversibly photoswitchable fluorophores to periodic dual color illumination, which will include thorough analysis in both adimensional and dimensioned spaces.

4. The term “radial frequency” is confusing. It should be “modulation frequency”.

In the revised manuscript, we replaced “radial frequency” by “angular frequency of modulation” (and not modulation frequency which could be confused with the inverse of the modulation period).

5. The phrase “without any cross talk” is repeated numerous times in the text. This is generally an incorrect and non-quantitative statement. It would be much better if the authors discuss cross talk in terms of a reduction factor (which can be very large but is still nonzero).

We replaced “without any cross talk” by “contrast enhancement” (with quantitative evaluations) in the revised manuscript.

6. The authors should add some discussion on crosstalk through detector saturation and through enhanced noise. In fact, all modulation based techniques are limited by these. Even a general discussion on the maximal magnitude of signal to background and on the practical number of independent channels realizable would do.

In the revised manuscript, we evaluated the Speed OPIOM limitations arising from noise considerations (with a measurement introduced in the last section of the Supporting Information) and from the limited precision on the phase to retrieve the quadrature delayed component of the collected images. Both types of limitations are now mentioned in the discussion of the revised Main Text together with an estimate on the practical number of independent channels we expect to be realizable with Speed OPIOM.

7. It would be good if the authors added to the SI a graph including all the emission spectra of the fluorescent proteins used in this work.

The revised SI includes a new figure displaying the emission spectra of the RSFPs, which have been used in this work.

REVIEWERS' COMMENTS:

Reviewer #2 (Remarks to the Author):

The authors have satisfactorily addressed issues raised by the referees. The manuscript can be accepted in its present form.